# Finite Volume Method for Modeling the Load-Rejection Process of a Hydropower Plant with an Air Cushion Surge Chamber

**Jianwei Lu** [1], **Guoying Wu** [2], **Ling Zhou** [1,3,*] and **Jinyuan Wu** [1,4]

1. College of Water Conservancy and Hydropower Engineering, Hohai University, Nanjing 210098, China
2. China Water Resources Pearl River Planning, Surveying and Designing Co., Ltd., Guangzhou 510610, China
3. Yangtze Institute for Conservation and Development, Nanjing 210098, China
4. Shanghai Municipal Engineering Design Institute (Group) Co., Ltd., Shanghai 200092, China
* Correspondence: zlhhu@163.com

**Abstract:** The pipe systems of hydropower plants are complex and feature special pipe types and various devices. When the Method of Characteristics (MOC) is used, interpolation or wave velocity adjustment is required, which may introduce calculation errors. The second-order Finite Volume Method (FVM) was presented to simulate water hammer and the load-rejection process of a hydropower plant with an air cushion surge chamber, which has rarely been considered before. First, the governing equations were discretized by FVM and the flux was calculated by a Riemann solver. A MINMOD slope limiter was introduced to avoid false oscillation caused by data reconstruction. The virtual boundary strategy was proposed to simply and effectively handle the complicated boundary problems between the pipe and the various devices, and to unify the internal pipeline and boundary calculations. FVM results were compared with MOC results, exact solutions, and measured values, and the sensitivity analysis was conducted. When the Courant number was equal to 1, the results of FVM and MOC were consistent with the exact solution. When the Courant number was less than 1, compared with MOC, the second-order FVM results were more accurate with less numerical dissipation. As the Courant number gradually decreased, the second-order FVM simulations were more stable. For the given numerical accuracy, second-order FVM had higher computational efficiency. The simulations of load rejection showed that compared with the MOC results, the second-order FVM calculations were closer to the measured values. For hydropower plants with complex pipe systems, wave velocity or the Courant number should be adjusted during MOC calculation, resulting in calculation error, and the error value is related to the parameters of the air cushion surge chamber (initial water depth, air cushion height, etc.). The second-order FVM can more accurately, stably, and efficiently simulate the load-rejection process of hydropower plants compared with MOC.

**Keywords:** air cushion surge chamber; method of characteristics; finite volume method; load rejection process

## 1. Introduction

In hydropower plants, complicated hydraulic transients often occur during startup, shutdown, or load change of the power generation unit. Dangerous water hammer events are caused by some inappropriate operations in the water system and likely produce abnormally high pressures, which may induce pipe rupture and damage other hydraulic devices. Due to the advantages of low construction cost and ecological environment impact, air cushion surge chambers have been widely used in water hammer protection in hydropower stations, ensuring the safety of the hydraulic operation [1]. However, water hammer protection devices also increase the operational complexity of the hydraulic system. Therefore, accurate and efficient numerical simulations of water hammer events become more important for the proper design and safe operation of hydropower plants.

The Method of Characteristics (MOC) is widely used in the simulation of hydraulic transition processes in hydropower plants [2]. However, in the actual complex water transmission system, there are certain short pipes, T-pipes, and tandem pipes. In the calculation of simulation with MOC modeling, interpolation calculation or wave speed adjustment is required; the former reduces the calculation efficiency and calculation accuracy and the latter often introduces calculation errors, which may lead to poor simulation results [1–3].

In recent years, many scholars have tried to use the Finite Volume Method (FVM) for the simulation of water hammer problems. Based on the system's mass and energy conservation, FVM can solve the discontinuous problem well without causing spurious numerical oscillations. Guinot [4] was the first to introduce the first-order FVM numerical method into the solution of the water hammer problem, and its calculation results are basically consistent with those of the MOC calculation. Zhao [5] developed the first-order and second-order FVM Godunov-type scheme (GTS) to simulate water-hammer problems in a simple horizontal pipeline with an instantaneously closed valve.

Zhou et al. [6,7] firstly developed a GTS approach to simulate transient cavitating pipe flow. Elong [8] solved the two-dimensional shallow water equations using the first-order finite volume method (FVM), the Harten Lax and van Leer (HLL) scheme, and the monotone upwind scheme for conservation laws (MUSCL) to simulate floods. Zhou [9] and Xue [10] conducted a simulation study of water–gas two-phase homogeneous flow using the FVM format and found that second-order FVM can effectively avoid spurious numerical oscillations and can improve the stability and accuracy of the calculation results.

Zhou et al. [2,3] developed one explicit solution source item approach for second-order FVM GTS to easily incorporate various forms of the existing unsteady friction models, including original convolution-based models, simplified convolution-based models, and Brunone instantaneous acceleration-based models. They pointed out that the first-order Godunov scheme and fixed-grid MOC scheme can achieve the same accuracy, but both display strong numerical damping once the Courant number is less than one. In contrast, the second-order GTS is more robust, even for Courant numbers significantly less than those for simple water hammer events.

Overall, the motivation and reason of attempting the second-order FVM GTS to simulate the load-rejection process of hydropower plants with an air cushion surge chamber are as follows. The fixed-grid MOC scheme is of first-order accuracy and has been widely used for solving water hammer equations in the simulation of hydraulic transition processes in hydropower plants. Since the real water pipe systems are usually complicated and made of pipe sections with different lengths, diameters, and materials, it is impossible to make the Courant number exactly equal to one in every pipe of such a complex pipe system; thus, the MOC scheme has to be implemented either via interpolation or wavespeed adjustment, which may induce large accumulated numerical errors. Importantly, as the Courant number is less than one and decreases, compared with MOC, the second-order GTS results are more accurate and more stable with less numerical dissipation. The previous work mainly focuses on the FVM GTS simulating the water hammer problem in a simple reservoir–pipe–valve system. However, it is necessary to further investigate the feasibility of GTS for more complicated hydraulic transient problems in a real hydraulic system with more complicated pipe components and devices, and to explore the possible computation error caused by the classic MOC scheme in a hydropower plant with a complex pipe system.

The main aim of this paper was to develop an accurate and efficient water hammer numerical model, which is significant for the proper design and safe operation of hydropower plants. A second-order FVM GTS fully considering the various pipe components and devices was developed to simulate the hydraulic transients and load-rejection process of the hydropower plant with an air cushion surge chamber, which has rarely been involved in previous published works. Importantly, the virtual boundary strategy was proposed to simply and effectively handle the complicated boundary problems between the pipe and various devices. Namely, virtual boundaries were introduced at the upstream and downstream boundaries and at the connection between the hydraulic components (air chamber

and unit) and the pipeline to achieve uniformity in the calculation of the control cells inside the pipeline and at the boundaries. The results calculated by the proposed second-order FVM GTS models were compared with the exact solution and the measured values as well as predictions by the MOC scheme. The accuracy and efficiency of the proposed approach were discussed. Another important purpose is that the proposed accurate model was used to explore the possible computation error caused by the MOC scheme in a hydropower plant with a complex pipe system.

## 2. Numerical Models of Hydropower Plant Hydraulic Transients

### 2.1. Water Hammer Control Equations

The classic water hammer equations for pipe flow are as follows [1]:

$$g\frac{\partial H}{\partial x} + V\frac{\partial V}{\partial x} + \frac{\partial V}{\partial t} + (J - gS_0) = 0 \tag{1}$$

$$V\frac{\partial H}{\partial x} + \frac{\partial H}{\partial t} + \frac{a^2}{g}\frac{\partial V}{\partial x} = 0 \tag{2}$$

where $H$ is the piezometric head; $V$ is the average flow velocity; $a$ is the water hammer wave speed; $g$ is the gravitational acceleration; $x$ is the coordinate distance along the tube axis; $t$ is time; $J$ is the steady-state friction coefficient of the pipe; and $S_0$ is the slope of the pipe.

### 2.2. Control Equations of Air Cushion Surge Chamber

In the analysis of air chamber shown in Figure 1, the pressure at any instant was assumed to be the same throughout the volume. The compressibility of the water in the air chamber was considered negligible compared with air compressibility. Inertia and friction were neglected. The air was assumed to follow the reversible polytropic relation

$$H_a V_a^k = H_{a0} V_{a0}^k = Constant \tag{3}$$

where $H_a$ and $V_a$ are the absolute pressure head and volume of the air within the air chamber, and their initial values are $H_{a0}$ and $V_{a0}$, respectively; and $k$ is the polytropic exponent. The adiabatic process with $k = 1.4$ is often used for the fast transients whereas the isothermal with $k = 1.0$ is often presented for the slower compressions; an intermediate polytropic case with $k = 1.2$ is often suggested as a reasonable compromise.

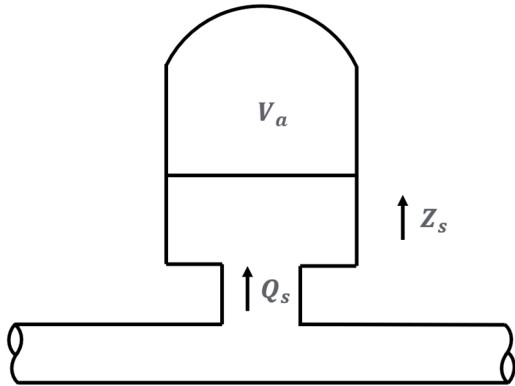

**Figure 1.** Schematic diagram of the air cushion surge chamber in the hydropower station.

The compressibility of the water and wall within the air chamber was neglected. $H_a$ is the absolute head equal to the gage plus barometric pressure heads

$$H_a = H_P - Z_s + H_{atm} - \left(R_S + \frac{1}{2gA_S^2}\right)|Q_S|Q_S \tag{4}$$

where $H_P$ is the piezometric head at the bottom of air chamber; $Z_s$ is the elevation of the air–water interface in air chamber; $H_{atm}$ is the absolute barometric pressure head; $Q_s$ is the inflow rate to the air chamber; $R_s$ is the head loss coefficient of the impedance hole of the air chamber; $A_s$ is the cross-section area of the air chamber.

The air volume was allowed to vary for inflow to and outflow from air chamber. The integrated continuity equation $dV_a/dt = -Q_s$, can be written as

$$Q_t = Q - Q_S = Q - A_S \frac{dZ_s}{dt} \tag{5}$$

where $Q$ and $Q_t$ are the flow rates at the inlet and outlet pipe of the bottom of the air chamber.

The piezometric head at the bottom of the air chamber, $H_P$, is associated with the inflow rate to the air chamber $Q_s$, which can be expressed as

$$\left. \begin{array}{c} H_P = C_2 - C_1 Q_s \\ C_1 = \left( \frac{1}{\left( \frac{1}{B_{P1}} \right) + \left( \frac{1}{B_{M2}} \right)} \right); C_2 = C_1 \left( \left( \frac{C_{P1}}{B_{P1}} \right) + \left( \frac{C_{M2}}{B_{M2}} \right) \right) \end{array} \right\} \tag{6}$$

where $C_{P1}, B_{P1}, C_{M2}, B_{M2}$ are boundary parameters that can be calculated from pressure heads and flow rates at the upstream and downstream pipe of the air chamber, and are discussed in the section concerning boundary treatment.

After combining Equations (3)–(6), the pressure head, flow rate, and water level at the air chamber can be obtained.

### 2.3. Control Equations of Hydraulic Turbine

The unit characteristic curve of the hydraulic turbine consists of the flow characteristic curve and the moment characteristic curve. Using modified Suter transformation [11], the flow function and torque function of the hydraulic turbine are as follows:

$$WH(x,y) = \frac{1}{\left( \frac{Q_{11}}{Q_{11r}} + c \right)^2 + \left( \frac{N_{11}}{N_{11r}} \right)^2} = \frac{h}{\left( q + c\sqrt{h} \right)^2 + n^2} \tag{7}$$

$$WB(x,y) = \frac{M_{11}}{M_{11r}} = \frac{m}{h} \tag{8}$$

$$\begin{cases} x = arctan\left[ \left( q + c\sqrt{h} \right)/n \right], & n \geq 0; \\ x = arctan\left[ \left( q + c\sqrt{h} \right)/n \right] + \pi, & n < 0; \end{cases} \tag{9}$$

where $x$ is the relative flow angle; $y$ is the relative guide vane opening; $WH(x,y)$ represents the flow functions; $WB(x,y)$ is the torque function; $Q_{11}$ is the unit flow rate; $Q_{11r}$ is the unit flow rate at rated operating conditions; $q = Q_{11}/Q_{11r}$ is the relative unit flow; $N_{11}$ is the unit speed; $N_{11r}$ is the unit speed at rated operating conditions; $n = N_{11}/N_{11r}$ is the relative unit speed; $H$ is the water head pressure; $H_r$ is the head pressure at rated operating conditions; $h = H/H_r$ is the relative head; $M_{11}$ is the unit torque; $M_{11r}$ is the unit torque at rated operating conditions; $m = M_{11}/M_{11r}$ is the relative unit moment; and the subscripts 11 and 11*r* indicate the unit value and the rated value, respectively.

When load rejection occurs, the unit speed equation is as follows:

$$n = n_0 + \frac{\Delta t}{T_a} (1.5m_0 - m_{00}) \tag{10}$$

where $T_a$ is the unit inertia time constant; $T_a = \frac{GD^2 N_r^2}{365 P_r}$; $GD^2$ is the unit rotational moment of inertia; $P_r$ is rated power output; $N_r$ is the rated speed; and the subscripts "0" and "00" indicate the first one time step and the first two time steps of the calculation time step, respectively.

The head balance equation [11,12] is given by

$$h = [C_{P1} - C_{M2} - (B_{P1} + B_{M2})Q_r q + C_3|q|q]/H_r \tag{11}$$

where coefficient $C_3 = Q_r^2 \left( \frac{1}{2gA_1^2} - \frac{1}{2gA_2^2} \right)$; $A_1$ is the inlet cross-sectional area of the worm shell; and $A_2$ is the outlet cross-sectional area of the tailpipe. Combining Equations (7), (8), (10) and (11), the head, flow rate, speed, and torque of the unit can be calculated.

## 3. Numerical Solution by Using the Second-Order Finite Volume Method

The matrix form of the water hammer equations (Equations (1) and (2)) can be expressed as follows:

$$\frac{\partial U}{\partial t} + A\frac{\partial U}{\partial x} = S \tag{12}$$

where $U = \binom{H}{V}$, $A = \begin{pmatrix} V & a^2/g \\ g & V \end{pmatrix}$, $S = \binom{0}{gS_0-J}$.

For the pipe water hammer problem, the Mach number is small, so the effect of the convective term can be neglected. The classical water hammer equation can be obtained by solving Equation (3) with the Riemann problem solution method.

$$\frac{\partial U}{\partial t} + \frac{\partial F}{\partial x} = S \tag{13}$$

where $F = \bar{A}U, \bar{A} = \begin{pmatrix} 0 & a^2/g \\ g & 0 \end{pmatrix}$.

The finite volume method was used to discretize the computational domain in the $x$-axis and $t$-axis, as shown in Figure 2, to form multiple computational control volumes with the fixed-grid length $\Delta x$ and then compute the control volumes.

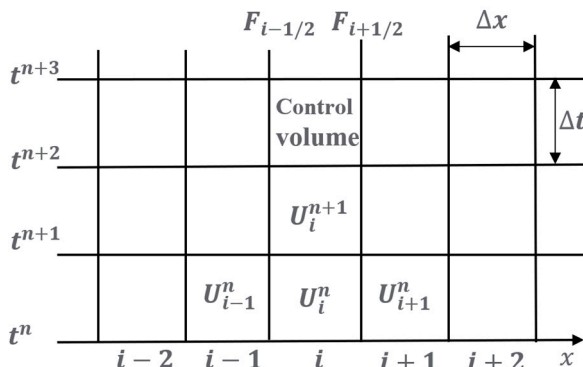

**Figure 2.** Grid system in the FVM Godunov scheme.

For the $i$th control volume, the integration of Equation (13) between control interfaces $i - 1/2$ and $i + 1/2$ yields

$$\mathbf{U}_i^{n+1} = \mathbf{U}_i^n - \frac{\Delta t}{\Delta x}\left(\mathbf{f}_{i+1/2}^n - \mathbf{f}_{i-1/2}^n\right) + \frac{\Delta t}{\Delta x}\int_{i-1/2}^{i+1/2} \mathbf{s}dx \tag{14}$$

where $\mathbf{U}_i$ is the average value of $\mathbf{u}$ within $[i - 1/2, i + 1/2]$; the superscripts $n$ and $n + 1$ indicate the $t$ and $t + \Delta t$ time levels, respectively; and $\mathbf{f}_{i+1/2}$ and $\mathbf{f}_{i-1/2}$ are the mass and momentum fluxes at the control interfaces $i - 1/2$ and $i + 1/2$, which are determined by solving a local Riemann problem at each cell interface [2,3].

### 3.1. Computation of Flux Term

In the Godunov approach, the numerical flux is determined by solving a local Riemann problem at each cell interface [4]. Applying Rankine–Hugoniot conditions $\Delta \mathbf{f} = \overline{A}\Delta \mathbf{u} = \overline{\lambda}_i \Delta \mathbf{u}$ where the eigenvalues $\overline{\lambda}_1 = -a$ and $\overline{\lambda}_2 = a$, the fluxes at $i + 1/2$ for all internal nodes and for $t \in [t^n, t^{n+1}]$ can be calculated by

$$\mathbf{f}_{i+1/2} = \overline{\mathbf{A}}_{i+1/2}\boldsymbol{u}_{i+1/2} = \frac{1}{2}\overline{\mathbf{A}}_{i+1/2}\left\{ \begin{pmatrix} 1 & a/g \\ g/a & 1 \end{pmatrix}\mathbf{U}_L^n - \begin{pmatrix} -1 & a/g \\ g/a & -1 \end{pmatrix}\mathbf{U}_R^n \right\} \tag{15}$$

in which $\overline{\mathbf{A}}_{i+1/2} = \mathbf{A}$; $\mathbf{U}_L^n =$ average value of $\mathbf{u}$ to the left of interface $i + 1/2$ at time $n$; and $\mathbf{U}_R^n =$ average value of $\mathbf{u}$ to the right of interface $i + 1/2$ at time $n$.

The estimation approach of $\mathbf{U}_L^n$ and $\mathbf{U}_R^n$ determines the accuracy order of the numerical scheme. In the first-order accuracy, $\mathbf{U}_L^n = \mathbf{U}_i^n$ and $\mathbf{U}_R^n = \mathbf{U}_{i+1}^n$. Herein, the MUSCL–Hancock method is used to achieve second-order accuracy in space and time, while the MINMOD limiter is suggested to avoid the spurious oscillations. The details of the MUSCL-Hancock method and the MINMOD limiter can also be found in a reference book (Toro 2009) [12].

The MUSCL–Hancock approach achieves a second-order extension of the Godunov scheme if the intercell flux $\mathbf{f}_{i+1/2}$ is computed according to the following steps [6]:

Step (1): Data Reconstruction. The data cell average values $\mathbf{U}_i^n$ are locally replaced by piece-wise linear functions in each cell $[x_{i-1/2}, x_{i+1/2}]$, and $\mathbf{U}_i^n$ at the extreme points are,

$$\mathbf{U}_i^L = \mathbf{U}_i^n - \frac{\Delta x}{2}\Delta_i, \ \mathbf{U}_i^R = \mathbf{U}_i^n + \frac{\Delta x}{2}\Delta_i \tag{16}$$

where $\Delta_i$ is a suitably chosen slope vector. The MINMOD limiter was used here to increase the order of accuracy of a scheme while avoiding spurious oscillations. Namely,

$$\Delta_i = \text{MINMOD}\left(\sigma_i^n, \sigma_{i-1}^n\right) = \begin{cases} \sigma_i^n, if, |\sigma_i^n| < |\sigma_{i-1}^n|, and, \sigma_i^n\sigma_{i-1}^n > 0 \\ \sigma_{i-1}^n, if, |\sigma_i^n| > |\sigma_{i-1}^n|, and, \sigma_i^n\sigma_{i-1}^n > 0 \\ 0, if, \sigma_i^n\sigma_{i-1}^n < 0 \end{cases} \tag{17}$$

where $\sigma_i^n = \left(\mathbf{U}_{i+1}^n - \mathbf{U}_i^n\right)/\Delta x$ and $\sigma_{i-1}^n = \left(\mathbf{U}_i^n - \mathbf{U}_{i-1}^n\right)/\Delta x$.

Step (2): Evolution. For each cell $[x_{i-1/2}, x_{i+1/2}]$, the boundary extrapolated values $\mathbf{U}_i^L$, $\mathbf{U}_i^R$ in Equation (16) are evolved by a time $0.5\Delta t$ according to

$$\begin{aligned} \overline{\mathbf{U}}_i^L &= \mathbf{U}_i^L + \frac{1}{2}\frac{\Delta t}{\Delta x}\left[\mathbf{f}\left(\mathbf{U}_i^L\right) - \mathbf{f}\left(\mathbf{U}_i^R\right)\right], \\ \overline{\mathbf{U}}_i^R &= \mathbf{U}_i^R + \frac{1}{2}\frac{\Delta t}{\Delta x}\left[\mathbf{f}\left(\mathbf{U}_i^L\right) - \mathbf{f}\left(\mathbf{U}_i^R\right)\right] \end{aligned} \tag{18}$$

Step (3): The Riemann Problem. To compute intercell flux $\mathbf{f}_{i+1/2}$, the conventional Riemann problem with data can be calculated by

$$\mathbf{U}_L^n \equiv \overline{\mathbf{U}}_i^R, \mathbf{U}_R^n \equiv \overline{\mathbf{U}}_{i+1}^L \tag{19}$$

Insert Equation (19) into Equation (15) and a second-order scheme for flux terms at $i + 1/2$ for all internal cell and for $t = [t^n, t^{n+1}]$ is obtained.

### 3.2. Incorporation of Source Term

When considering the pipe friction resistance, the second-order Runge–Kutta solution is used to obtain the second-order calculation accuracy explicit results, and the calculation process is as follows.

First step:

$$\overline{\boldsymbol{u}}_i^{n+1} = \mathbf{U}_i^n - \frac{\Delta t}{\Delta x}\left(\mathbf{f}_{i+(1/2)}^n - \mathbf{f}_{i-(1/2)}^n\right) \tag{20}$$

Second step:

$$\overline{\overline{u}}_i^{n+1} = \overline{u}_i^{n+1} + \frac{\Delta t}{2} s\left(\overline{u}_i^{n+1}\right) \tag{21}$$

Last step:

$$u_i^{n+1} = \overline{u}_i^{n+1} + \Delta t s\left(\overline{\overline{u}}_i^{n+1}\right) \tag{22}$$

The time step should satisfy the CFL convergence condition [1,11], i.e.,

$$Cr = \frac{a\Delta t}{\Delta x} \leq 1 \tag{23}$$

$$N = \frac{L}{\Delta x} \tag{24}$$

where $Cr$ is the Courant number; $N$ is the number of pipe grids; and $L$ is the pipe length.

For the water hammer problem, the Courant number actually refers to the relative relationship between the time step $\Delta t$ and the space step $\Delta x$ [1,12]. When $Cr$ is greater than 1, the calculation result is unstable; when $Cr$ is less than 1, and the closer to 0, the more serious the numerical dissipation is, and the accuracy of the calculation result is worse. Therefore, the range of the Courant number is $Cr$ less than or equal to 1, and preferably equal to 1 or close to 1.

### 3.3. Virtual-Boundary Strategy

Boundary conditions including the interior device boundary in the hydraulic system of the hydropower station contain the upstream head-constant reservoir, air chamber, turbine, and the downstream reservoir. As discussed above, in the second-order Godunov scheme, the head and flow rate of the $i$th control volume at time $t + \Delta t$ are calculated by combining the parameters of the upstream two (($i - 2$)th, ($i - 1$)th) and the downstream two (($i + 1$)th, ($i + 2$)th) control volumes at time $t$. Therefore, numerically, boundary conditions are expected to provide numerical fluxes $\mathbf{f}_{1/2}$, $\mathbf{f}_{N+1/2}$, which are required in order to update the extreme cells $I_1$ and $I_N$ to the next time level $n + 1$.

In this paper, as shown in Figure 3, virtual control volumes $I_{-1}$ and $I_0$ adjacent to $I_1$ and virtual control volumes $I_{N+1}$ and $I_{N+2}$ adjacent to $I_N$ were proposed to realize second-order Godunov scheme at the upstream and downstream control volumes of the computational domain, respectively. The corresponding fluxes $\mathbf{f}_{1/2}$ and $\mathbf{f}_{N+1/2}$ were computed in the same method as the interior control volumes.

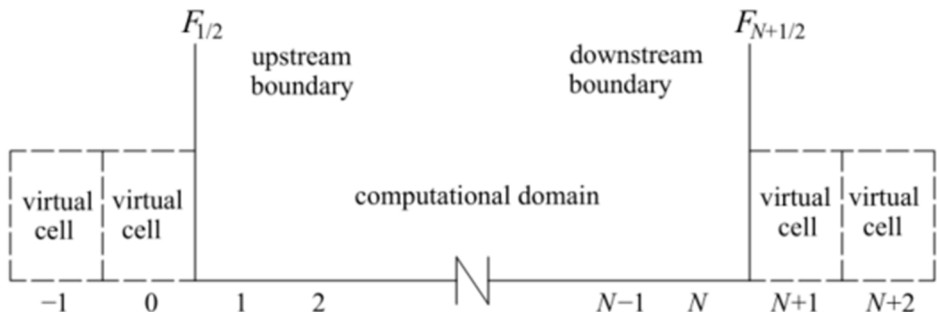

**Figure 3.** Grid diagram introducing virtual cells at the boundaries of reservoir and hydropower unit.

The head and flow rate in the virtual control volumes were assumed to be identical with those at boundaries, namely

$$\mathbf{U}_{-1}^{n+1} = \mathbf{U}_0^{n+1} = \mathbf{U}_{1/2} = \begin{pmatrix} H_{1/2} \\ V_{1/2} \end{pmatrix} \tag{25}$$

$$\mathbf{U}_{N+1}^{n+1} = \mathbf{U}_{N+2}^{n+1} = \mathbf{U}_{N+1/2} = \begin{pmatrix} H_{N+1/2} \\ V_{N+1/2} \end{pmatrix} \tag{26}$$

In the Godunov scheme, the Rankine–Hugoniot condition across each wave of speed $\bar{\lambda}_i$ gives the following relations,

$$\frac{a}{g}(V_{i+1/2} - V_R) - (H_{i+1/2} - H_R) = 0 \tag{27}$$

$$\frac{a}{g}(V_{i+1/2} - V_L) + (H_{i+1/2} - H_L) = 0 \tag{28}$$

At the upstream reservoir boundary, from the Riemann invariance equation (Equation (27)), it follows that

$$H_{1/2} - \frac{a}{g}V_{1/2} = H_1^n - \frac{a}{g}V_1^n \tag{29}$$

At the downstream reservoir boundary, from the Riemann invariance equation (Equation (28)), it follows that:

$$H_N^n + \frac{a}{g}V_N^n = H_{N+1/2} + \frac{a}{g}V_{N+1/2} \tag{30}$$

where $V_1^n$ and $H_1^n$ are the velocity and pressure head of the first control volume adjacent to the upstream reservoir; $V_N^n$ and $H_N^n$ are the velocity and pressure head of the last control volume adjacent to the downstream reservoir; and $H_{1/2}$ and $H_{N+1/2}$ are the head pressures of the upstream and downstream reservoirs, respectively.

For the upstream and downstream boundaries of the hydraulic turbine, from the turbine control equations, only the physical variable values of the virtual control volumes at the worm housing and at the tail pipe are required to derive the physical variable values at the turbine. Therefore, combining the Riemann invariance equations (Equations (27) and (28)), it is obtained

$$C_{P1} = H_N^n + \frac{a}{g}V_N^n \tag{31}$$

$$B_{P1} = \frac{a}{gA_1} \tag{32}$$

$$C_{M2} = H_1^n + \frac{a}{g}V_1^n \tag{33}$$

$$B_{M2} = \frac{a}{gA_2} \tag{34}$$

where $V_N^n$ and $H_N^n$ are the flow rate and head of the last control volume of the upstream pipe at the snail shell; and $V_1^n$ and $H_1^n$ are the flow rate and head of the first control volume of the downstream pipe at the right end of the tail pipe, respectively. The obtained Equations (31) to (34) are brought into Equation (11) to solve the head balance equation under virtual boundary conditions.

Similarly, for the upstream and downstream boundaries of air chamber, from the control equations of the air chamber, combining the Riemann invariance equations (Equations (27) and (28)), Equations (31)–(34) can be obtained and brought into Equation (6) to solve the head balance equation under virtual boundary conditions.

## 4. Numerical Solution by Using Method of Characteristics

The momentum and continuity Equations (1) and (2) are transformed into four ordinary differential equations by the MOC [1].

$$C^+ : \begin{cases} \frac{g}{a}\frac{dH}{dt} + \frac{dV}{dt} + \frac{fV|V|}{2D} = 0 \\ \frac{dx}{dt} = +a \end{cases} \tag{35}$$

$$C^- : \begin{cases} -\frac{g}{a}\frac{dH}{dt} + \frac{dV}{dt} + \frac{fV|V|}{2D} = 0 \\ \frac{dx}{dt} = -a \end{cases} \tag{36}$$

where $f$ is the Darcy–Weisbach friction factor; $D$ is pipe diameter.

As shown in Figure 4, integration of $C^+$ along characteristic lines from interior (fixed grid) point $A$ to point $P$, and integration of $C^-$ along characteristic lines from interior (fixed grid) point $B$ to point $P$, can be written as

$$C^+ : H_i^{n+1} = C_P - B_P Q_i^{n+1} \tag{37}$$

$$C^- : H_i^{n+1} = C_M + B_M Q_i^{n+1} \tag{38}$$

in which $Q$ is the flow rate; and the coefficients $C_P$, $B_P$, $C_M$, and $B_M$ are known constants when the equations are applied.

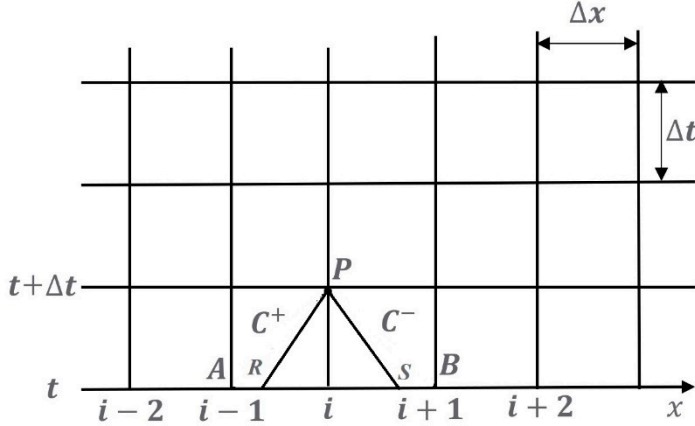

**Figure 4.** Definition of the sketch in the $x$–$t$ plane with a fixed MOC grid for the water hammer problem.

When the Courant number $Cr \leq 1$ ($Cr = \frac{a\Delta t}{\Delta x}$), the space–line interpolation fixed-grid MOC scheme can give the values of the coefficients $C_P$, $B_P$, $C_M$, and $B_M$ along the $C^+$ and $C^-$ characteristic lines as follows:

$$C_P = H_{PR} + B{\cdot}Q_{PR} \tag{39}$$

$$B_P = B + R{\cdot}Cr{\cdot}|Q_{PR}| \tag{40}$$

$$C_M = H_{PS} - B{\cdot}Q_{PS} \tag{41}$$

$$B_M = B + R{\cdot}Cr{\cdot}|Q_{PS}| \tag{42}$$

in which, $B$ is a function of the physical properties of the fluid and the pipeline, often called the pipeline characteristic impedance, and $B = a/gA$, A is the cross-section area; $R$ is the pipeline resistance coefficient $R = f\Delta x/(2gDA^2)$; as shown in Figure 4, $Q_{PR}$ and $H_{PR}$ are the flow rate and pressure head at $R$ node; $Q_{PS}$ and $H_{PS}$ are the flow rate and pressure head at $S$ node; their values can be calculated by interpolation,

$$Q_{PR} = Q_i^n - Cr{\cdot}\left(Q_i^n - Q_{i-1}^n\right) \tag{43}$$

$$Q_{PS} = Q_i^n - Cr{\cdot}\left(Q_i^n - Q_{i+1}^n\right) \tag{44}$$

$$H_{PR} = H_i^n - Cr{\cdot}\left(H_i^n - H_{i-1}^n\right) \tag{45}$$

$$H_{PS} = H_i^n - Cr{\cdot}\left(H_i^n - H_{i+1}^n\right) \tag{46}$$

Combining Equations (39) and (40), $H_i^{n+1}$ and $Q_i^{n+1}$ at the interior node can be obtained. Similarly, the pressure head and flow rate of the boundary nodes adjacent to the hydropower unit can be calculated by combining Equation (39), the control equations of the hydraulic turbine, and Equation (40).

## 5. Results and Discussion

The main purposes of this section are (1) to investigate the accuracy, stability, and efficiency of second-order FVM GTS and MOC in a simple reservoir–pipe–valve system; (2) to validate the proposed second-order FVM model by comparing the calculated and measured data of load rejection in a hydropower plant with a complicated pipe system; (3) to explore the possible computation error caused by the MOC scheme in a complex pipe system of the hydropower plant; and (4) to study the effect of air chamber parameters on the error of MOC scheme simulating the hydraulic events in the hydropower plant.

### 5.1. Water Hammer Problem in a Simple Reservoir–Pipe–Valve System

The classical "reservoir–pipe–valve" system is used to verify the accuracy of the proposed. The upstream is a reservoir, and the downstream is a valve connected by a single pipe. The pipe is 800 m long, which is divided into 16 control volumes. The water hammer wave velocity is 1000 m/s, and the upstream reservoir head is 20 m. The initial velocity of the pipe is 0.15 m/s. The water hammer problem is caused by the instantaneous valve closure. It is assumed that the pipe wall is smooth, which means any dissipation is caused by the numerical calculation.

Figures 5 and 6 show the water hammer solutions for the simple system using MOC and second-order FVM, respectively, to investigate the effect of different Courant numbers $Cr$ (1.0, 0.7, 0.5, 0.3, 0.1) on the calculation results of the two solution schemes. The accuracy and efficiency of FVM and MOC water hammer calculations were analyzed.

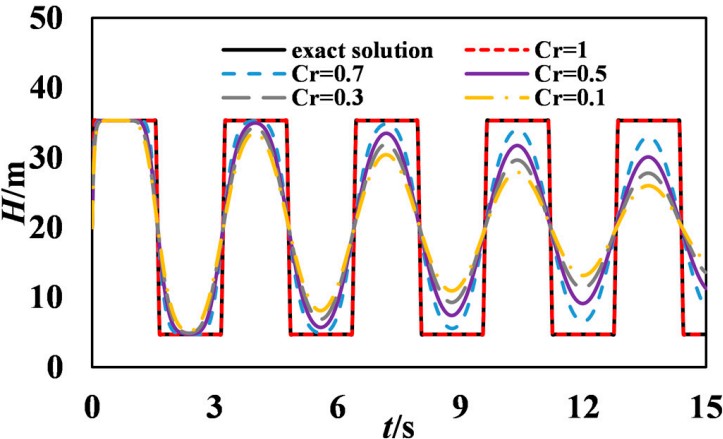

**Figure 5.** Pressure head calculated by the MOC scheme with different Courant numbers.

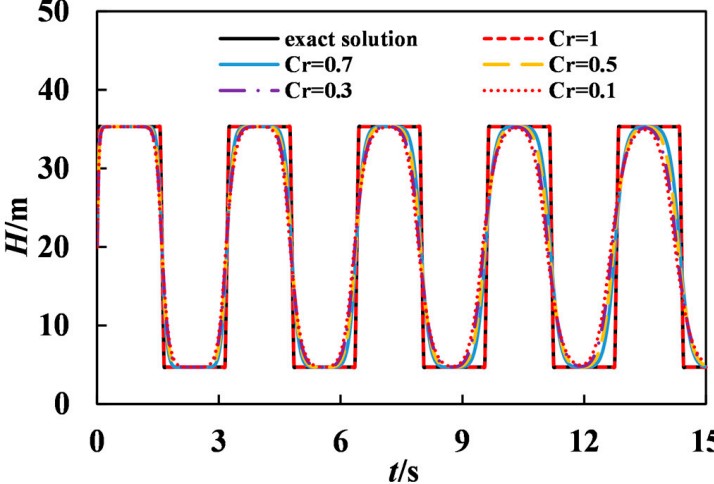

**Figure 6.** Pressure head calculated by second-order FVM with different Courant numbers.

The results in Figures 5 and 6 indicate that when $Cr = 1.0$, the results of both FVM and MOC calculations were identical with the exact solution (i.e., the analytical solution is obtained by the analytical method when the Courant number was equal to 1). When $Cr < 1.0$, both computational results had numerical dissipation. For the same $Cr$, the numerical dissipation of MOC was more severe, e.g., for $Cr = 0.1$, the initial energy (peak pressure) of MOC was dissipated by 26% in 15 s, while the FVM in the second-order Godunov scheme was only dissipated by 1.06%.

Figure 7 shows that when $Cr < 1.0$, the second-order FVM was more stable and less dissipative than the MOC scheme for the same number of control volumes ($N_S = 32$). At $Cr = 0.3$, in order to reach the same numerical accuracy, MOC needed $N_S = 256$, while only $N_S = 32$ was used in the second-order FVM scheme. Table 1 displays that the for the same computation accuracy, the computation time in the MOC scheme (0.19 s with $N_S = 256$) was about 5 times that in the second-order FVM scheme (0.037 s, $N_S = 32$). Therefore, when $Cr < 1.0$, the second-order FVM scheme is more efficient than the MOC scheme for the same computation accuracy.

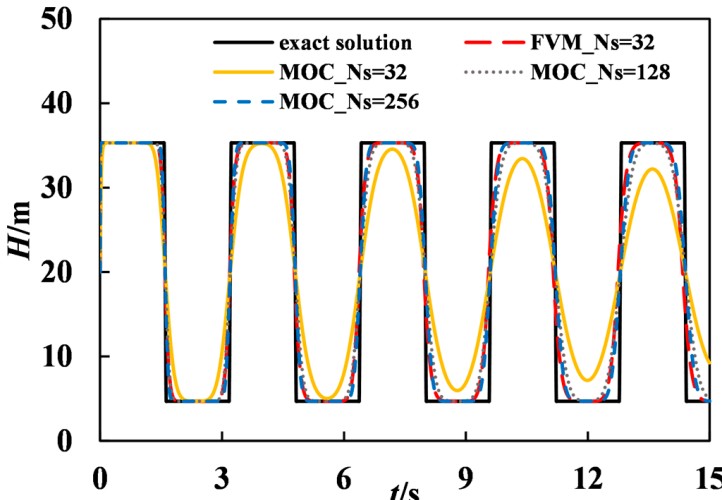

**Figure 7.** Comparison of pressure heads calculated by second-order FVM and MOC with different grid numbers at $Cr = 0.3$.

**Table 1.** Computation time of MOC and second-order FVM with different numbers of grids.

| Number of Grids | MOC Calculation Time/s | FVM Calculation Time/s |
|---|---|---|
| 32 | 0.012 | 0.037 |
| 128 | 0.069 | 0.555 |
| 256 | 0.19 | 1.849 |

Overall, for water hammers in a simple pipe, the second-order FVM model is accurate, efficient, and stable even for Courant numbers less than one. For the given Courant number and the same accuracy, the proposed model is far more efficient than the MOC model.

### 5.2. Hydraulic Transients in Actual Hydropower Plant

#### 5.2.1. Project Overview

One real hydropower station has two turbine units through branch pipes, one air chamber, and pressurized pipes between the upstream and downstream reservoirs. The layout of the water transmission system is shown in Figure 8. The pipe parameters are shown in Table 2, while the parameters of the turbine units are shown in Table 3.

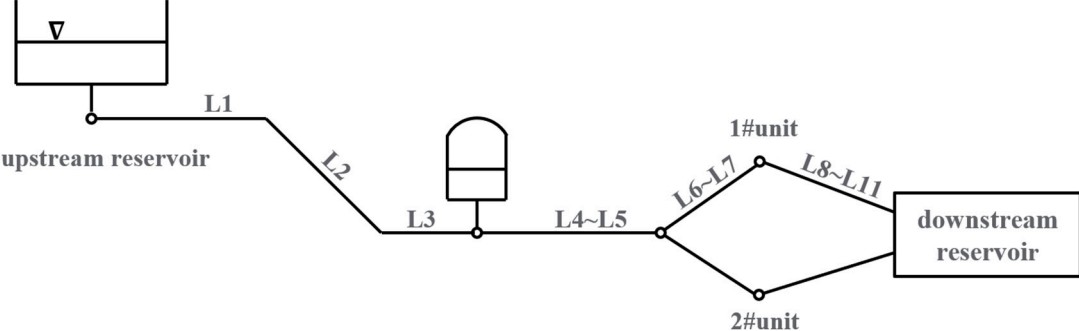

**Figure 8.** Schematic diagram of a hydropower plant layout with an air cushion chamber.

**Table 2.** Parameters of the water pipe system in the numerical simulations.

| Pipe Number | Pipe Length/m | Wave Speed/(m·s⁻¹) | Roughness |
|---|---|---|---|
| L1 | 15.39 | 976.4 | 0.014 |
| L2 | 169.26 | 976.4 | 0.014 |
| L3 | 20.77 | 976.4 | 0.014 |
| L4 | 56.4 | 976.4 | 0.015 |
| L5 | 26.6 | 976.4 | 0.014 |
| L6 | 100.33 | 1202.3 | 0.013 |
| L7 | 5.4 | 1210.8 | 0.013 |
| L8 | 14 | 1045.1 | 0.014 |
| L9 | 70.94 | 1045.1 | 0.014 |
| L10 | 25.52 | 1152.75 | 0.014 |
| L11 | 13.6 | 1152.75 | 0.014 |

**Table 3.** Unit parameters of the water turbine in the numerical simulations.

| Unit Parameters | Numerical Value |
|---|---|
| Single machine capacity (MW) | 150 |
| Rated head (m) | 105.8 |
| Rated flow rate (m³·s⁻¹) | 148.8 |
| Rated speed (r·min⁻¹) | 200 |
| Power Rating (kW) | 139,000 |
| Rotational inertia (t·m²) | 10,920 |

In this section, two field experiment cases on load rejection are introduced to investigate the accuracy of the numerical models.

Field Experiment Case A of Load Rejection: The water levels of upstream and downstream reservoirs are 412.4 m and 290.97 m, respectively. The guide vane closing law adopts "two stages": initial guide vane opening 73.8%, first closing time 3.62 s, second closing time 32.53 s (closing to 10% of no load), inflection point guide vane opening 60%, guide vane inactivity time 0.27, and total closing time 43.04 s. The turbine operating parameters are based on the rated parameters of the unit.

Field Experiment Case B of Load Rejection: The water levels of upstream and downstream reservoirs are 406.08 m and 290.6 m, respectively. The law of guide vane closing adopts "two-stage": initial guide vane opening 74.3%, the first stage closing time is 5.64 s, the second stage closing time is 30.53 s, and the inflection point guide vane opening is 61.13%. The total closing time is 36.17 s. The rated parameters of the turbine are used for the operating parameters.

5.2.2. Comparison to Field Experimental Data

When MOC is used to model the complicated pipe system, there are often two treatment methods, including (1) MOC (Scheme 1) being used to adjust the wave speed so that $Cr = 1$, and (2) MOC (Scheme 2) being used to keep the wave speed invariant and increase

the number of pipe grids so that *Cr* is as close to 1 as possible. According to the above two schemes, the adjusted wave speed *a*, number of grids *N*, and Courant number *Cr* are shown in Table 4.

**Table 4.** Wave speed, grid number, and Courant number in pipe sections in the MOC simulations.

| Pipe Number | MOC (Scheme 1) | | | MOC (Scheme 2) | | |
|---|---|---|---|---|---|---|
| | $a$/(m·s$^{-1}$) | $N$ | $Cr$ | $a$/(m·s$^{-1}$) | $N$ | $Cr$ |
| L1 | 961.875 | 4 | 1.000 | 976.400 | 31 | 0.983 |
| L2 | 984.070 | 43 | 1.000 | 976.400 | 346 | 0.998 |
| L3 | 1038.500 | 5 | 1.000 | 976.400 | 42 | 0.987 |
| L4 | 1007.143 | 14 | 1.000 | 976.400 | 115 | 0.995 |
| L5 | 950.000 | 7 | 1.000 | 976.400 | 54 | 0.991 |
| L6 | 1194.405 | 21 | 1.000 | 1202.300 | 166 | 0.995 |
| L7 | 1350.000 | 1 | 1.000 | 1210.800 | 8 | 0.897 |
| L8 | 1166.667 | 3 | 1.000 | 1045.100 | 26 | 0.970 |
| L9 | 1043.235 | 17 | 1.000 | 1045.100 | 133 | 0.980 |
| L10 | 1063.333 | 6 | 1.000 | 1152.750 | 44 | 0.994 |
| L11 | 1133.333 | 3 | 1.000 | 1152.750 | 23 | 0.975 |

As discussed in Section 5.1, it is clear that the second-order FVM still maintains high computational accuracy even when *Cr* is less than 1. Therefore, for complex pipe components and devices of the hydropower plant, the characteristics of each pipe section (pipe length, wave speed, etc.) keep invariance in the second-order FVM simulation, and only the Courant number and the number of pipe section grids need to be adjusted. The number of pipe section grids is determined as follows.

In the calculation of the second-order FVM, to ensure the stability of the calculation, the pipe grid must satisfy the Courant condition, taking the *i*th pipe as an example, i.e.,

$$C_{r_i} = \frac{a_i \Delta t}{\Delta x_i} \leq 1 \tag{47}$$

where in order to ensure the synchronization of the calculation at all pipe sections, the calculation time step $\Delta t$ for each pipe section is the same; $\Delta x_i$ is the grid length of the *i*th pipe section, m; $a_i$ is the wave speed of the *i*th pipe section, m·s$^{-1}$; $C_{r_i}$ is the Courant number of the *i*th pipe section.

In the FVM calculation, the grid number $N_i$ is calculated by the following equation.

$$N_i = \frac{L_i}{\Delta x_i} \tag{48}$$

where $N_i$ is the grid number of the *i*th pipe; $L_i$ is the length of the *i*th pipe, m.

Substitute Equation (36) into Equation (35) to obtain the following equation

$$N_i = C_{r_i} \frac{L_i}{a_i \Delta t} \tag{49}$$

From Equation (37), it can be seen that in order to ensure that $N_i$ is an integer, it is necessary to adjust the Courant number *Cr* of the *i*th pipe, and the principle of adjustment is as follows: the range of the Courant number is less than or equal to 1, and preferably equal to 1 or close to 1. Using the abovementioned method, the wave speed *a*, grid number *N*, and *Cr* of each segment in the second-order FVM calculation can be obtained, as shown in Table 5.

**Table 5.** Wave speed, grid number, and Courant number in pipe sections in the FVM simulations.

| Pipe Number | FVM | | |
| --- | --- | --- | --- |
| | $a$/(m·s$^{-1}$) | $N$ | $Cr$ |
| L1 | 976.400 | 3 | 0.761 |
| L2 | 976.400 | 43 | 0.992 |
| L3 | 976.400 | 5 | 0.940 |
| L4 | 976.400 | 14 | 0.969 |
| L5 | 976.400 | 6 | 0.881 |
| L6 | 1202.300 | 20 | 0.959 |
| L7 | 1210.800 | 1 | 0.897 |
| L8 | 1045.100 | 3 | 0.896 |
| L9 | 1045.100 | 16 | 0.943 |
| L10 | 1152.750 | 5 | 0.903 |
| L11 | 1152.750 | 2 | 0.678 |

The maximum rotational speed during load rejection is an important index of hydraulic transient control. For the two field experiment cases on load rejection, the simulation results of MOC (Scheme 1), MOC (Scheme 2), and second-order FVM are given to compare the experimental rotational speed in Table 6 and Figures 9a and 10a, and the corresponding transient pressures at the worm gear are displayed in Figures 9b and 10b.

**Table 6.** Maximum rotational speed during 100% load rejection in two field experiment cases and the calculation results of MOC and FVM.

| Experiment Case | Experimental Rotational Speed (r·min$^{-1}$) | FVM (r·min$^{-1}$) | MOC (Scheme 1) (r·min$^{-1}$) | MOC (Scheme 2) (r·min$^{-1}$) |
| --- | --- | --- | --- | --- |
| A | 283.84 | 282.416 | 281.455 | 280.946 |
| B | 279.2 | 279.076 | 278.081 | 277.672 |

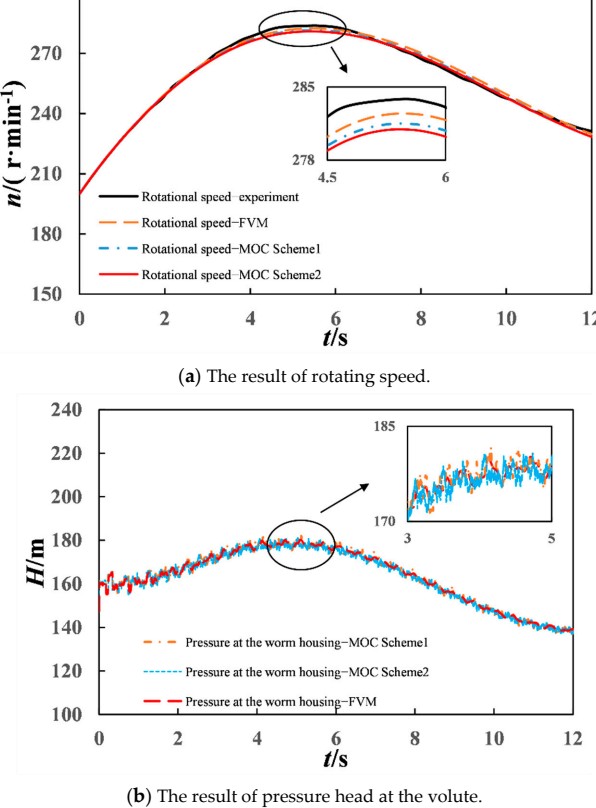

(**a**) The result of rotating speed.

(**b**) The result of pressure head at the volute.

**Figure 9.** Comparison of the calculation results (FVM and MOC) and the measured data in Field Experiment Case A of Load Rejection.

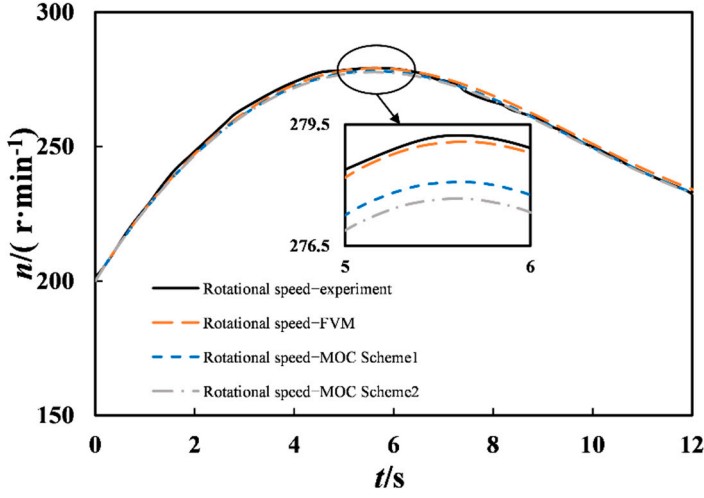

(**a**) The result of rotating speed.

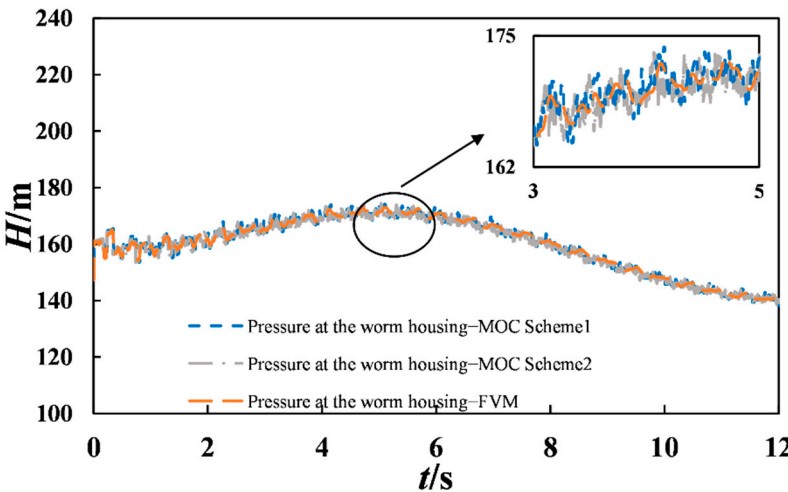

(**b**) The result of pressure head at the volute.

**Figure 10.** Comparison of the calculation results (FVM and MOC) and the measured data in Field Experiment Case B of Load Rejection.

Figures 9a and 10a show that the MOC (Scheme 1), MOC (Scheme 2), and FVM simulation results of unit rotation speed during 100% load rejection basically matched with the experimental results. However, the second-order FVM model better reproduced the experimental data, and the simulation results were more accurate than those of MOC. The reason for the larger calculation error caused by MOC is that MOC adjusts the water hammer wave speed or reduces the Courant number. Compared with MOC (Scheme 2), the simulation results of MOC (Scheme 1) were slightly better. The reason for this result is that the Courant number Cr in MOC (Scheme 2) in each pipe section was less than one, which led to more serious dissipation of the MOC (Scheme 2) than that of the MOC (Scheme 1) with slight wave speed adjustment. However, FVM does not need to adjust the wave velocity of the pipe, and only needs to reduce the Cr condition appropriately. Compared with MOC, FVM not only simplified the simulation process, but also had better calculation accuracy.

As shown in Figures 9b and 10b, for the simulation of the pressure at the worm gear, the FVM simulation results were more stable with less fluctuation than those of both MOC (Scheme 1) and MOC (Scheme 2).

### 5.2.3. Effect of Air Chamber Parameters on the Error of MOC Scheme

The abovementioned results demonstrated that the second-order FVM method proposed in this paper could perfectly reproduce the exact solution and the field experimental data in the simple pipe system and the real complex pipe system. In this section, the second-order Godunov FVM simulation results are taken as the benchmark to study the effect of air chamber parameters on the error of MOC (Scheme 1).

A.    The effect of static water depth in design condition

Here, the air chamber control constant method $CT_0$ is used, keeping $P_0 \cdot V_0$ constant. When the cross-section area of air chamber remains unchanged, the $P_0 \cdot l_0$ value can be treated constant, in which $l_0$ is the air length of air chamber. So, the static water depth $L_{s0}$ under design condition can be derived from the total height of the air chamber and air length. In this paper, five static water depths $L_{s0}$ were selected to study their effects on 100% load rejection hydraulic transients of the power station. The MOC and FVM simulation results are shown in Table 7 and Figure 11.

**Table 7.** Comparison of the maximum rotational speeds calculated by FVM and MOC with differently designed air chamber water depths.

| Design Water Depth(m) | Maximum Rotational Speed (FVM) $(r \cdot min^{-1})$ | Maximum Rotational Speed (MOC) $(r \cdot min^{-1})$ |
|---|---|---|
| 4.8 | 278.068 | 277.068 |
| 5.4 | 278.55 | 277.555 |
| 6 | 279.076 | 278.081 |
| 6.6 | 279.626 | 278.629 |
| 7.2 | 280.154 | 279.148 |

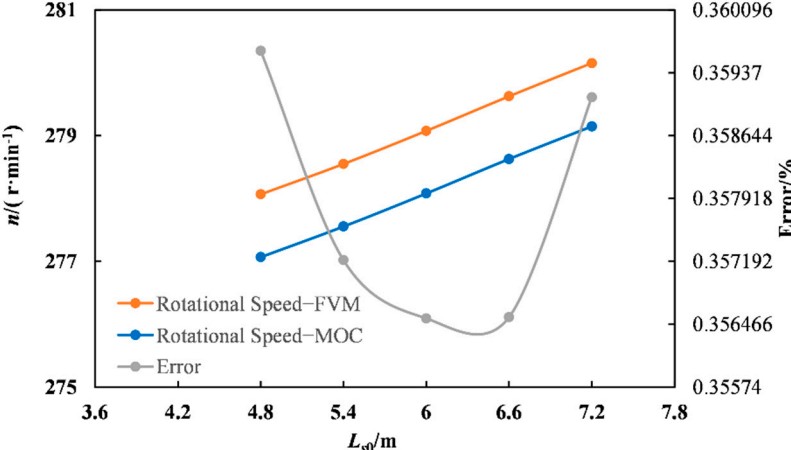

**Figure 11.** The maximum rotational speeds calculated by FVM and MOC with different design air-chamber water depths and the MOC computation error.

As shown in Table 7 and Figure 11, with the increase of $L_{s0}$, the maximum rotational speed gradually increased. The error of MOC calculation had a trend of decreasing and then increasing, with a slight change between 0.35% and 0.36%.

B.    The effect of air cushion height

Under the given values of the design air pressure and design water depth, as the roof elevation of the air chamber increased, the air cushion height (air length) $l_0$ increased. Five air cushion height (air length) $l_0$ were selected to study their effects on 100% load rejection hydraulic transients of the power station. The MOC and FVM simulation results are shown in Table 8 and Figure 12.

**Table 8.** Comparison of the maximum rotational speeds calculated by FVM and MOC with different air cushion heights.

| Air Cushion Height (m) | Maximum Rotational Speed (FVM) (r·min$^{-1}$) | Maximum Rotational Speed (MOC) (r·min$^{-1}$) |
|---|---|---|
| 4 | 279.076 | 278.081 |
| 5 | 278.202 | 277.203 |
| 6 | 277.459 | 276.451 |
| 7 | 279.626 | 275.836 |
| 8 | 276.363 | 275.336 |

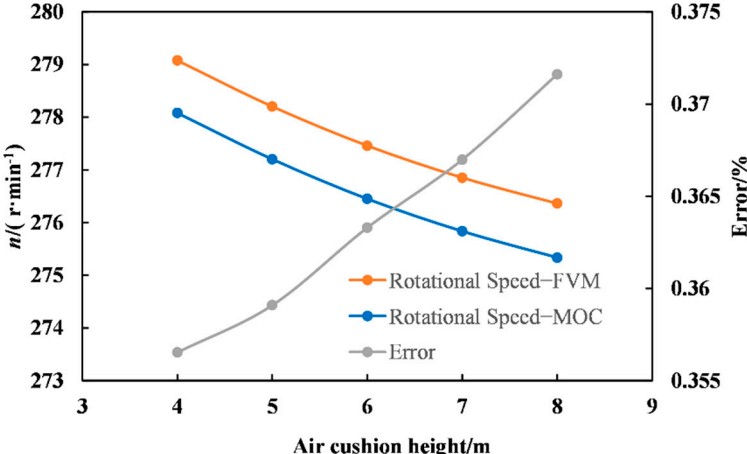

**Figure 12.** The maximum rotational speeds calculated by FVM and MOC with different air cushion heights and the MOC computation error.

As shown in Table 8 and Figure 12, with the increase of air cushion height (air length) $l_0$, the maximum rotational speed gradually decreased. The error of MOC calculation had a trend of decreasing, with change between 0.355% and 0.375%. It indicates that the simulation effect of MOC became worse with the increase of air cushion height. For high head hydropower plants, when the air cushion height is large, it is advisable to use FVM for simulation in order to ensure the calculation accuracy.

C.    Effect of polytropic exponent $k$

The thermodynamic process of the closed air chamber was between isothermal and isentropic, and polytropic exponent $k$ ranged from 1.0 to 1.4, which is the range commonly recognized and adopted at present [1]. Five polytropic exponents $k$ were selected to study their effects on 100% load rejection hydraulic transients of the power station. The MOC and FVM simulation results are shown in Table 9 and Figure 13.

**Table 9.** Comparison of the maximum rotational speeds calculated by FVM and MOC with different polytropic exponents.

| Polytropic Exponent | Maximum Rotational Speed (FVM) (r·min$^{-1}$) | Maximum Rotational Speed (MOC) (r·min$^{-1}$) |
|---|---|---|
| 1 | 279.077 | 278.082 |
| 1.1 | 279.077 | 278.081 |
| 1.2 | 279.076 | 278.084 |
| 1.3 | 279.076 | 278.082 |
| 1.4 | 276.076 | 278.081 |

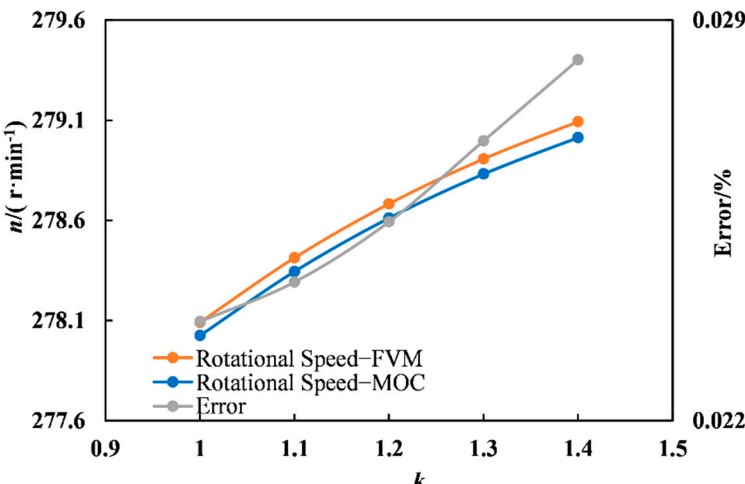

**Figure 13.** The maximum rotational speeds calculated by FVM and MOC with different polytropic exponents and the MOC computation error.

As shown in Table 9 and Figure 13, with the increase of polytropic exponent $k$, the maximum rotational speed gradually increased. The error of MOC calculation had a trend of increasing, with slight change between 0.022% and 0.029%.

## 6. Conclusions

In this paper, the second-order FVM Godunov scheme model was developed to simulate hydraulic transients and load rejection in a hydropower plant with an air chamber. The virtual boundary strategy was proposed to simply and effectively handle the complicated boundary problems. The results of the proposed model were compared with MOC results, the exact solution, and the measured data. The main conclusions are as follows.

(1) The second-order FVM Godunov scheme model can more accurately, stably, and efficiently simulate the water hammer problem in pipe systems. When the Courant number was $Cr = 1$, both calculated results of FVM and MOC were consistent with the exact solution. When the Courant number was $Cr < 1$, both computational results had numerical dissipation. As the Courant number gradually decreased, the second-order FVM simulation results were more stable. For the given Courant number, the numerical dissipation of MOC was more serious. The second-order FVM is more efficient for the same accuracy.

(2) For the load-rejection process of hydropower units containing an air chamber, the results calculated by the proposed FVM model were basically consistent with the measured rotational speed variation, which verifies that the second-order FVM model can be accurate for the simulation analysis of load-rejection process of hydropower units containing complex pipe systems.

(3) For complex pipe systems, the second-order FVM model better reproduced the experimental data, and the simulation results were more accurate than those of MOC. The reason for the larger calculation error caused by MOC is that MOC adjusts the water hammer wave speed or reduces the Courant number. The second-order FVM does not need to adjust the wave velocity of the pipe, and only needs to reduce the $Cr$ condition appropriately. Compared with MOC, FVM not only simplifies the simulation process, but also has better calculation accuracy.

(4) The error of MOC calculation is associated with the air chamber parameters. For the current case of hydropower plant, with the increase of static water depth in the design condition, the error of MOC calculation had a trend of decreasing and then increasing, with a slight change between 0.35% and 0.36%. With the increase of air cushion height (air length), the error of MOC calculation had a trend of decreasing, with a change between 0.355% and 0.375%. With the increase of polytropic exponent $k$, the error of MOC calculation had a trend of increasing, with a slight change between 0.022% and 0.029%.

Overall, the second-order FVM model was robust in simulating the water hammer problems in a simple or complex pipe system. Considering the higher accuracy, stability, and efficiency, the high-order FVM is feasible and suggested for water hammer simulation in real hydraulic systems with more complicated pipe components and devices.

**Author Contributions:** Writing—original draft preparation, J.L., G.W. and J.W.; resources, writing, review and editing, L.Z. All authors have read and agreed to the published version of the manuscript.

**Funding:** This research was funded by National Natural Science Foundation of China, grant numbers 51839008 and 51679066.

**Data Availability Statement:** Some or all data, models, or code that support the findings of this study are available from the corresponding author upon reasonable request.

**Conflicts of Interest:** The authors declare no conflict of interest.

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
