# Peer review of "Finite Volume Method for Modeling the Load-Rejection Process of a Hydropower Plant with an Air Cushion Surge Chamber"

_water, doi:10.3390/w15040682_

Round 1
Reviewer 1 Report
The work in this paper all appears to have been carefully done, and is certainly relevant to the scope of Water. It is therefore suitable for publication. The only comment I would make is that the authors should go through all the figure captions (and also table captions) and make them a bit more descriptive of what is actually being presented. Captions should be more than just 6-7 words! The rest of the paper is also a bit technical at times, and would likely be difficult for non-experts to follow. So if possible maybe add a bit more explanation here and there, if you want to make it more accessible and interesting to non-experts? For experts in this area though it should be clearly understandable as it is already.
Reviewer 2 Report
This study focuses on the finite volume method for modeling load rejection process of hydropower plant with an air cushion surge chamber. The content includes equations, decomposition of the equations, test of accuracy, stability and efficiency of the method. Compared MOC method (both scheme 1 and scheme 2) with FVM method, discussion on the influence of Cr number. However, to my understanding, there is no need to use second-order accuracy FVM to solve equation (12) since this is a first-order equation and a theoretical solution could be given. Also, could you please show (1) the notation of W in equation (7), and (2) all coefficients in the calculation: q, n, h, m, Ta. (3) One mesh example, (4) a brief description of the principle of MOC. MOC scheme 2 needs more grids and results are not satisfied compared with scheme 1, could you provide some reasons? I think this manuscript is acceptable after major revision.
Round 2
Reviewer 2 Report
Accept in the revised form